# Tim-3 is dispensable for allergic inflammation and respiratory tolerance in experimental asthma

Carolin Boehne[1], Ann-Kathrin Behrendt[1¤a], Almut Meyer-Bahlburg[1,2¤a], Martin Boettcher[3¤b], Sebastian Drube[3], Thomas Kamradt[3], Gesine Hansen[1,2,4]*

**1** Department of Pediatrics and Adolescent Medicine, Pediatric Pulmonology, Allergology and Neonatology, Hannover Medical School, Lower Saxony, Germany, **2** Biomedical Research in Endstage and Obstructive Lung Disease Hannover (BREATH), Member of the German Center for Lung Research (DZL), Hannover Medical School, Lower Saxony, Germany, **3** Institute of Immunology, University Hospital Jena, Jena, Thuringia, Germany, **4** Cluster of Excellence RESIST (EXC 2155), Hannover Medical School, Lower Saxony, Germany

¤a Current address: Department of Pediatrics, University Medicine Greifswald, Greifswald, Mecklenburg-Western Pomerania, Germany
¤b Current address: Department of Medicine, Hematology and Oncolgy, University of Erlangen, Erlangen, Bavaria, Germany
* Hansen.Gesine@mh-hannover.de

**Data Availability Statement:** All relevant data are within the paper and its Supporting Information files.

## Abstract

T cell immunoglobulin and mucin domain-containing molecule-3 (Tim-3) has been described as a transmembrane protein, expressed on the surface of various T cells as well as different cells of innate immunity. It has since been associated with Th1 mediated autoimmune diseases and transplantation tolerance studies, thereby indicating a possible role of this receptor in counter-regulation of Th2 immune responses. In the present study we therefore directly examined the role of Tim-3 in allergic inflammation and respiratory tolerance. First, Tim-3$^{-/-}$ mice and wild type controls were immunized and challenged with the model allergen ovalbumin (OVA) to induce an asthma-like phenotype. Analysis of cell numbers and distribution in the bronchoalveolar lavage (BAL) fluid as well as lung histology in H&E stained lung sections demonstrated a comparable degree of eosinophilic inflammation in both mouse strains. Th2 cytokine production in restimulated cell culture supernatants and serum IgE and IgG levels were equally increased in both genotypes. In addition, cell proliferation and the distribution of different T cell subsets were comparable. Moreover, analysis of both mouse strains in our respiratory tolerance model, where mucosal application of the model allergen before immunization, prevents the development of an asthma-like phenotype, revealed no differences in any of the parameters mentioned above. The current study demonstrates that Tim-3 is dispensable not only for the development of allergic inflammation but also for induction of respiratory tolerance in mice in an OVA-based model.

**Funding:** The authors received no specific funding for this work.

**Competing interests:** The authors have declared that no competing interests exist.

## Introduction

Within the last decades the prevalence of allergic asthma has risen steadily in western countries affecting more than 300 million people worldwide by now. It is a very common chronic lung disease that is characterized by eosinophilic airway inflammation, airway hyperreactivity and reversible airway obstruction [1]. Type-2 high and low forms of airway inflammation are discriminated, taking into account the multifactorial pathogenesis of asthma. About 50% of patients are characterized by a type- 2 high immune status involving innate as well as adaptive immune cells [2]. The latter is defined by a Th2 dominated cell environment insufficiently counterbalanced by Th1 as well as Treg cells [3, 4]. We and others have highlighted the role of T cell surface expressed proteins such as costimulatory or coinhibitory molecules in the modulation of allergic immune reaction [5–11].

Tim-3 is a member of the T cell immunoglobulin and mucin domain-containing (Tim) family of genes (Tim-1-8), of which Tim-1, -3 and -4 are conserved in humans [12]. Based on the original description of Tim-3 as a transmembrane protein, primarily expressed on CD4$^+$ Th1 effector cells but not on Th2 cells, it is well known for its role as an important inhibitory molecular checkpoint [13]. (Auto)-inflammatory as well as dysfunctional and exhaustive cell responses are ascribed to its expression and function on Th17, Treg and CD8$^+$ T cells [14] though references for an activating role exist as well [15]. Besides its inhibitory function in adaptive immunity Tim-3 also plays a role in proinflammatory cytokine production [16, 17] and is frequently expressed on innate immune cells such as macrophages, dendritic cells (DC), natural killer (NK) cells, monocytes and mast cells [14, 16]. A variety of autoimmune disorders [13, 18–21], chronic viral infections [22, 23] and different cancers [14] have since been linked to Tim-3.

The variable influence of Tim-3 on adaptive Th1 as well as innate immunity and its location within the T cell and airway phenotype regulator (*Tapr*) locus in mice have raised the question whether Tim-3 gene polymorphisms are linked with allergy [24–27]. To date, some murine studies investigated a putative role of Tim-3 in allergic airway disease with partly controversial results: use of anti-Tim-3 antibodies [28] or small interfering RNA [29], respectively, lead to a significant reduction of airway hyperreactivity in murine OVA-based models, whereas further studies stated that Tim-3 is not essential for a type-2 driven allergic response [30, 31]. However, recent human studies revealed an upregulation of Tim-3 on CD4$^+$ T cells in patients with allergic asthma thereby suggesting a possible association with Th1/Th2 imbalance [32, 33].

Given the diverse picture of Tim-3 phenotype in innate and adaptive immunity, in both the human system and murine models so far, we investigated Tim-3$^{-/-}$ mice in our well-established murine model of allergic lung disease and respiratory tolerance [34, 35]. Systemic sensitization and local challenge of Tim-3$^{-/-}$ mice with the model allergen OVA resulted in the development of an allergic phenotype. Moreover, we characterized the development of respiratory tolerance by mucosal OVA pre-treatment prior to sensitization (tolerization) in both Tim-3$^{-/-}$ and WT mice and described the functional T cell environment in Tim-3 deficient mice.

## Materials and methods

### Mice

12-16 week old BALB/cByJ female mice were purchased from Elevage Janvier (France). Tim-3$^{-/-}$ BALB/c mice were bred in our animal facility. In brief, the gene encoding for Tim-3 (*Havcr2*) was replaced via gene targeting in embryonic stem cells. Homologue recombination led to deletion of exons 1 and 2 of the Tim-3 gene which was identified via PCR and flow cytometry of peripheral blood cells. Chimeric mice from Tim-3 deficient embryonic stem cell

clones were then crossed to a BALB/c background for six to eight generations [19]. All mice were maintained under specific pathogen-free conditions with free access to food and water. Animal experiments were strictly performed according to institutional and state guidelines. To alleviate suffering during intraperitoneal (i.p.) and intranasal (i.n.) application of compounds, the mice received brief inhalation anaesthesia with isoflurane (100% v/v, Baxter, Germany). Mice were sacrificed by an overdose of isoflurane followed by blood loss during dissection. The local animal welfare committee of Lower Saxony State Office for Consumer Protection and Food Safety approved protocols (TSA 11/0950).

## Immunization protocols

To provoke an allergic inflammatory phenotype (*OVA*), WT and Tim-3$^{-/-}$ mice were immunized with OVA (Sigma-Aldrich, Germany) [7, 9, 34, 36]. Lipopolysaccharide (LPS) contaminations of OVA preparations (Sigma, grade V, LPS content $\geq$1500 EU/mg protein) were removed as previously described [36, 37]. Sensitization was accomplished by i.p. application of OVA twice– on days 0 and 14 – with OVA (20 µg in 200 µl 0.9% saline) adsorbed to 2 mg of an aqueous solution of aluminium hydroxide and magnesium hydroxide (Imject Alum; Perbio Science, Germany). Subsequently, i.n. challenges with 20 µg OVA in 40 µl 0.9% saline were performed on three consecutive days, day 14 to 16 and day 20 to 22 (Fig 1A). WT and Tim-3$^{-/-}$ control mice (*Alum*) received 200 µl 0.9% saline / 2 mg Imject Alum i.p. and were challenged i. n. with 0.9% saline. Mice were sacrificed on day 23.

I.n. mucosal application of high-dosed OVA prior to i.p. sensitization prevents the manifestation of an allergic phenotype in our model resulting in respiratory tolerance to the allergen (*TOL*). Therefore, WT and Tim-3$^{-/-}$ mice were pre-treated with OVA i.n. (500 µg OVA, 40 µl 0.9% saline) on days -6 and -3. Afterwards they were referred to the allergy protocol described above (Fig 1B).

## Collection and analysis of bronchoalveolar lavage fluid (BALF)

During dissection of mice the trachea was cannulated and whole lung was inflated once with 0.8 ml of ice-cold PBS supplemented with 2 mM EDTA. Subsequently, right lung volume was

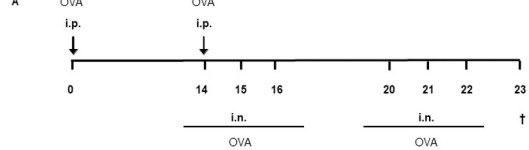

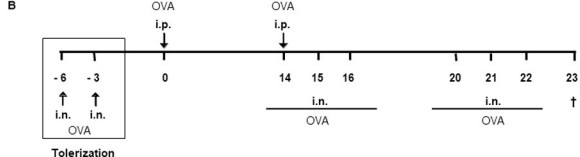

**Fig 1. Immunization protocols.** (A) Induction of allergic phenotype. WT and Tim-3$^{-/-}$ mice were sensitized at days 0 and 14 by -i.p. application of OVA in Alum, resulting in an allergic phenotype; whereas negative control mice were i.p. immunized with NaCl 0.9% in Alum. Sensitization was followed by i.n. challenges at consecutive days 14-16 and 20-22 with either OVA or saline, respectively. Section of all mice at day 23. (B) Induction of tolerant phenotype. In order to induce a tolerant phenotype, mice received i.n. tolerization with OVA in saline at days -6 and -3 (square), followed by i.p. immunization with OVA in Alum at days 0 and 14 and i.n. challenges with OVA at consecutive days 14-16 and 20-22. Section of all mice at day 23.

aspirated, followed by three consecutive lavages (instillation and subsequent aspiration of 0.4 ml of ice-cold PBS/EDTA) until a total BALF volume of 1.6 ml per mouse was reached. Total BALF volume was centrifuged at 300 x g (4°C, 10 minutes) and BALF cellular constituents were resuspended in 0.5 ml PBS/EDTA. Total number of BALF cells was afterwards determined using an automated cell counting system based on Trypan Blue dye exclusion method (Cedex HiRes, Innovatis, Germany). In addition, BALF differentials were determined on cytospins ($1x10^5$ BALF cells per cytospin) subjected to May-Grünwald-Giemsa staining. At least 100 cells were differentiated by light microscopy based on overall morphologic criteria, including cell size and shape of nuclei by a blinded examiner. Total BALF leukocyte subsets were quantified by multiplication of percent values with the respective total BALF cell counts.

## Analysis and score-based quantification of lung histology

Left lungs were inflated with 0.4 ml PBS/EDTA and fixed in 4% formalin at room temperature for at least 24 hours. Fixed lungs were then embedded in paraffin, deparaffinized, cut into 4 μm-thick sections and stained with haematoxylin and eosin (H&E, Merck, Germany) or with periodic acid–schiff reagent (PAS, Sigma, Germany). Evaluation and quantification of the degree of pulmonary inflammation (H&E) as well as mucus production (PAS) was done under masked conditions on lung histology slides, scanned with a digital camera (40×; 10 shots per lung) and analyzed with HistoClick Software (provided by I. Hohmann and W. Scherf, Germany), based on morphometric image analysis developed in our laboratory. Positive pixel counts were first totalled for each individual lung and then averaged per group of mice. The pixel data were finally presented as mean ± SEM per group. Exemplary lung sections of individual mice were scanned with a digital camera (x200) for illustration of pulmonary inflammation or mucus production.

## Measurement of OVA-specific immunoglobulins IgE, IgG$_1$ and IgG$_{2a}$

Serum was obtained from each individual mouse 24 hours after the last i.n. OVA challenge. OVA-specific IgE, IgG$_1$ and IgG$_{2a}$ serum levels were measured by ELISA. Using a standard curve for mouse anti-OVA IgE (AbD Serotec, UK), we calculated OVA-specific IgE concentrations (pg/ml). Data for OVA-specific serum IgG$_1$ and IgG$_{2a}$ are presented as titre values derived from analysis of optical density values versus factors of serum dilution series with a logarithmic curve fitting model.

## Measurement of in vitro cytokine production and proliferation

Single cell suspensions from spleens were prepared and $5x10^6$ cells were cultured *in vitro* with OVA (0.2 mg/ml) in culture medium (RPMI medium supplemented with 10% FCS, 100 U/ml penicillin and 100 μg/ml streptomycin). Cytokines (IL-5, IL-10, IL-13, IFN-γ) were measured in supernatants of spleen cells at day 3 of restimulation with OVA by commercially available ELISA (R&D Systems, USA), according to the manufacturer's instructions. To determine proliferation, OVA-restimulated splenocytes were analyzed using a cell titre glow assay (Promega, Germany) according to the manufacturer's instructions. In detail, 0.1 ml cell suspension and 0.1 ml freshly prepared glow reagent were mixed and measured using a GloMax Multi-Detection System (Promega, Germany).

## Flow cytometry and antibodies (Abs)—Antigen-specific Th cells

Unless otherwise indicated, mAbs were grown, purified and conjugated from hybridoma supernatants in our laboratory. Splenocytes were isolated and counted. $1x10^6$ cells were

cultured in 1 ml culture medium in the presence of 3 μg/ml anti-CD28 (clone 37.51) and 50 μg/ml OVA (Sigma-Aldrich, Germany) for 6 hours at 37˚C and 5% $CO_2$. For the last 4 hours 5 μg/ml Brefeldin-A (Sigma Aldrich, Germany) was added. A pooled sample restimulated with 3 μg/ ml anti-CD28 and 3 μg /ml anti-CD3 (clone 145-2C11) served as a positive control. For a negative control pooled cells were restimulated with anti-CD28 alone.

After the restimulation cells were harvested and washed twice by centrifugation in cold PBS. Cells were labeled with LIVE/DEAD® Fixable Aqua Dead Cell Stain Kit (life Technologies, Germany) for later discrimination of live and dead cells according to the manufacturer's instructions. After an additional washing step in cold PBS, cells were fixed in 2% para-formaldehyde for 20 min at 4˚C. Next, cells were washed with PBA (PBS with 0,25% BSA and 0.02% sodium acetate), permeabilized with PBA-S (PBA with 0.5% saponine) and stained with the following antibodies for 30 min at 4˚C. For blocking of unspecific binding anti-CD16/CD32 (clone 2.4G2) and rat IgG (Jackson Immuno Research, USA) as well as fluorescently labeled antibodies anti-CD4 APC/Cy7 (ebioscience, Germany), anti-CD154 APC (miltenyi biotech, Germany), anti-IL-4 Alexa Fluor® 488 (ebioscience, Germany) and anti-IL-5 PE (clone TRFK4) were used. 1 000 000 events were acquired for each sample using a LSRII cell cytometer (BD Biosciences, Mountain View, USA). Data were analyzed using FlowJo Software (TreeStar, Ashland, USA).

## Flow cytometry and antibodies (Abs)–staining of CD8[+], CD4[+]and Foxp3[+] T cells

Splenic single-cell suspensions were incubated with fluorescently labeled Abs for 20 min at 4˚C in staining buffer (PBS with 0.5% BSA). Abs used included reagents specific for CD4 (RM4-5) and CD8 (53-6.7) from BD Pharmingen (San Diego, USA) and Foxp3 (FJK-16s) from eBioscience (San Diego, USA). Intracellular staining of Foxp3 was performed using the eBioscience reagents according to manufacturer's instructions as previously described [7]: after surface staining, cells were fixed for 45 min in fixation buffer. Subsequently, cells were washed twice with perm buffer and incubated with Ab to Foxp3 for 30 min at 4˚C. Data was collected on a FACSCanto flow cytometer (BD Biosciences, Mountain View, USA), analyzed using FlowJo software (Tree Star, Ashland, USA) and further statistically evaluated applying GraphPad Prism V software (La Jolla, USA).

## Statistical analysis

Statistical analyses were performed using GraphPad Prism V software (La Jolla, USA). Intergroup differences of cytokine levels in supernatants of spleen cells and proliferation were calculated using Student's unpaired $t$-test. For all other parameters a non-Gaussian distribution was assumed and significance levels among the groups were calculated using Mann-Whitney-$U$-test. $p$ values of less than 0.05 were considered significant (*); values of less than 0.01 were considered highly significant (**). Data are presented as mean ± SEM.

## Results

### Tim-3 is dispensable for the development of allergic airway inflammation in mice

Mice, immunized and challenged with the model allergen OVA according to the protocol described above, develop an allergic phenotype characterized by an eosinophilic airway inflammation, high allergen-specific serum IgE levels as well as increased Th2 cytokine production (*OVA*) compared to control animals (*Alum*) [7, 9, 34] (Figs 2 and 3). To evaluate the role of Tim-3 for the development of this allergic phenotype, Tim-3[-/-] mice and WT mice were systemically

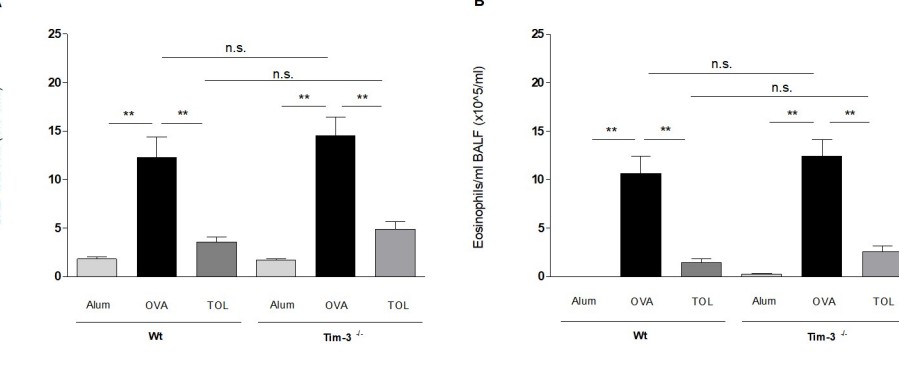

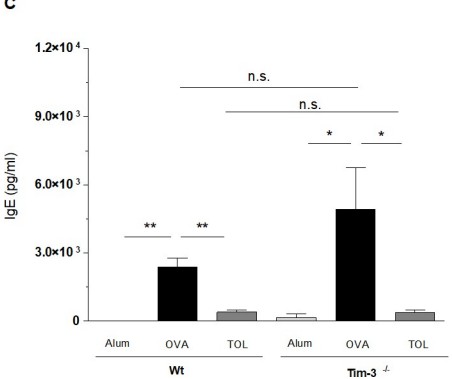

**Fig 2. BALF analysis and OVA-specific immunoglobulin E.** (A, B) Bronchoalveolar lavage fluid was obtained from each individual mouse (WT, Tim-3$^{-/-}$) at section day to determine total cell count (A) and number of eosinophils (B). No significant differences were detected between WT and Tim-3$^{-/-}$ *OVA* and *TOL* mice, resulting in comparable allergic and tolerant phenotypes. (Data of three independent experiments, n = 12-15 animals per group.) (C) Significant and comparable decrease in IgE production for WT and Tim-3$^{-/-}$ *TOL* mice compared to *OVA* controls. (One representative out of three independent experiments, n = 5-6 animals for each group.) Mann-Whitney-*U*-test. $^{*}$ $p \leq 0.05$; $^{**}$ $p \leq 0.01$. All data are presented as mean ± SEM.

immunized with the model allergen OVA in Alum and afterwards challenged intrapulmonary with OVA in NaCl 0,9%. Control mice received Alum and NaCl 0.9% instead. First, we compared the degree of lung inflammation in both mouse strains. Total cell numbers and differential counts in the BALF were comparable between Tim-3$^{-/-}$ and WT mice (Figs 2 and S1). Moreover, H&E stained lung sections revealed an equal degree of peribronchiolar and perivascular cell infiltrates. In parallel, PAS stained lung sections showed a comparable degree of mucus production and goblet cell hyperplasia. The comparability of lung inflammation and mucus hypersecretion in Tim-3$^{-/-}$ and WT mice could be confirmed by objective, investigator independent computer-based morphometric image analysis of H&E and PAS stained lung tissues (Fig 4).

Next, we examined the production of allergen-specific serum immunoglobulins in Tim-3$^{-/-}$ and WT mice after immunization to OVA (Figs 2C and S2). All three analyzed immunoglobulins, IgE, IgG$_1$ and IgG$_{2a}$, were equally modified by sensitization to OVA in the two different mouse strains, meaning a significant increase of IgE, IgG$_1$ and IgG$_{2a}$ compared to control mice. Our findings demonstrate that Tim-3 is not essential for the modification of the immunoglobulin response for the allergic phenotype in our murine model.

To analyze the effect of Tim-3 on cytokine production in response to OVA immunization, we next measured different cytokine levels in supernatants of spleen cell cultures in WT and Tim-3$^{-/-}$ mice after restimulation with OVA. As expected, we detected significantly elevated levels of IL-5, IL-10 and IL-13 in OVA-immunized WT mice compared to WT controls. In

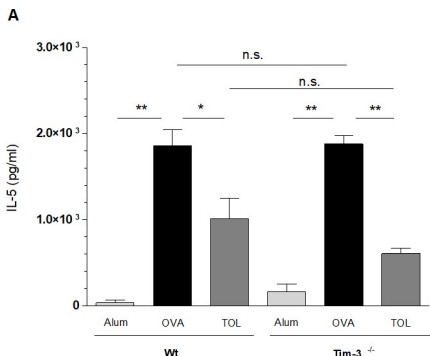
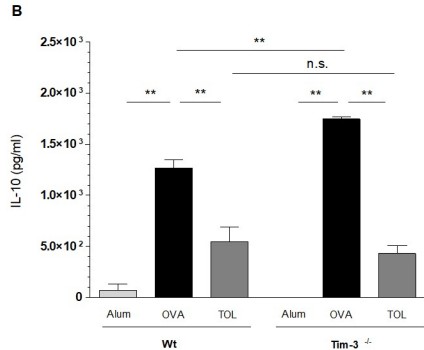

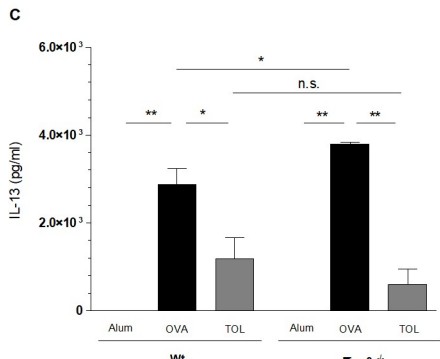
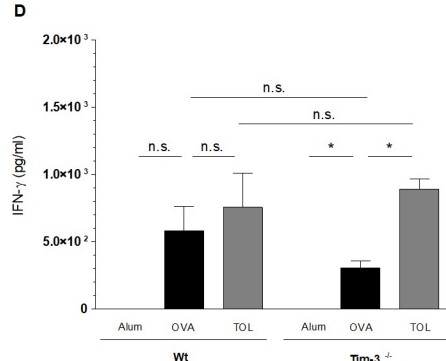

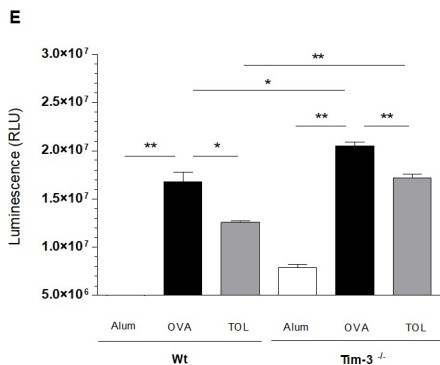

**Fig 3. OVA-specific cytokine production and proliferation of splenic lymphocytes.** (A-C) Production of IL-5 (A), IL-10 (B) and IL-13 (C) in cell culture supernatants of spleens showed a significant decrease after tolerization compared to allergic (*OVA*) animals in both, WT and Tim-3$^{-/-}$ mice. (D) Tolerant mice showed a significant increase of IFN-γ in Tim-3$^{-/-}$ mice. (E) Proliferation of splenic lymphocytes by cell titer glow assay after OVA restimulation showed decreased proliferation in *TOL* animals of WT and Tim-3$^{-/-}$ mice. Increased levels of luminescence were detected in allergic mice, both WT and Tim-3$^{-/-}$ animals, no significant differences between WT and Tim-3$^{-/-}$ mice. (One representative out of three independent experiments; n = 5-6 animals per group.) Student's unpaired t-test.* p ≤ 0.05, ** p ≤0.01, n.s. p > 0.05. All data are presented as mean ± SEM.

accordance to our findings for lung inflammation and immunoglobulin production, we found no differences for cytokine levels between OVA-immunized Tim-3$^{-/-}$ and WT mice (Fig 3A–

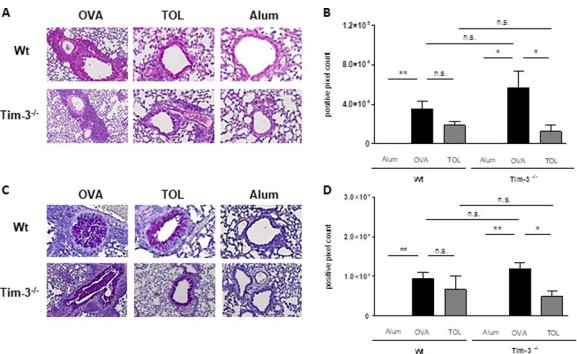

**Fig 4. Lung histology analysis.** (A-D) Lungs were stained for eosinophilic lung inflammation (H&E, A) or mucus secretion (PAS, C), followed by computer-based quantification using an image analyzing program (B and D). Significantly decreased inflammation and mucus production was found in *TOL* mice compared to allergic controls (*OVA*). In contrast, WT and Tim-3$^{-/-}$ mice were not significantly altered. (Three independent experiments; n = 5-6 animals per group.) Mann-Whitney-*U*-test. $^*$ p $\leq$ 0.05, $^{**}$ p $\leq$0.01, n.s. p > 0.05. All data are presented as mean ± SEM.

3C). Only the IFN-γ concentration was slightly lower in OVA immunized Tim-3$^{-/-}$ compared to WT mice, however, this difference did not reach the level of significance (Fig 3D). *Ex vivo* proliferation of splenocytes in response to restimulation with OVA assessed by cell titre glow assay showed a dose-dependent increase of proliferation in *OVA* mice compared to *Alum* controls in WT and Tim-3$^{-/-}$ mice. However, we found an overall increased basal proliferation in Tim-3$^{-/-}$ mice compared to WT mice (Fig 3E).

FACS analysis of antigen (Ag)-specific Th cells isolated from Tim-3$^{-/-}$ and WT mice after immunization to OVA showed that Ag-specific Th cells –defined by simultaneous staining of CD4 and CD154 (CD40L) - were similarly enhanced in frequency and number in both mouse strains (Fig 5). CD154 (CD40L) is a marker that is upregulated within a few hours after T cell receptor triggering and has been previously shown to be induced only in those T cells that have recognized their cognate Ag [38–41]. Moreover, the number of OVA-specific CD154$^+$CD4$^+$ IL-4 and IL-5 producing cells were similar in WT and Tim-3$^{-/-}$ mice (S3 Fig). In OVA-immunized mice the frequency of CD4$^+$CD154$^+$ OVA-specific cells was approximately 7% of the CD154$^+$ cells and of those about 75% produce IL-4 or IL-5 (S4 Fig).

## Tim-3 is dispensable for respiratory tolerance in a murine model

Next, we aimed at analyzing the role of Tim-3 for the development of respiratory tolerance and compared Tim-3$^{-/-}$ and WT mice in the corresponding murine model.

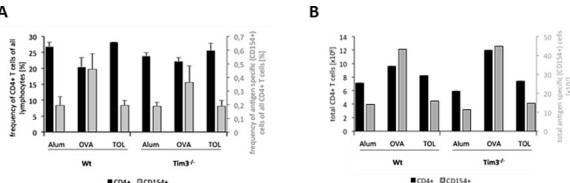

**Fig 5. Antigen-specific Th cells.** (A, B) WT or Tim-3$^{-/-}$ mice were immunized with OVA/Alum i.p. (*OVA*), PBS/Alum i.p. (*Alum*), or received OVA i.n. (*TOL*) before challenge with OVA i.n. as described in methods. 23 days later mice were sacrificed and single cell suspensions prepared from spleens were analyzed flow cytometrically. The frequency (A) and absolute numbers (B) of CD4$^+$ Th cells (black bars) and CD4$^+$CD154$^+$ cells (grey bars) were determined as described in methods. Number of animals per group was two. Error bars represent the standard deviation.

To induce respiratory tolerance, we applied a high dose of OVA i.n. prior to sensitization and i.n. challenges as described above. As shown in previous studies, mucosal application of the antigen prior to referral of mice to the asthma protocol inhibits the development of eosinophilic airway inflammation (Figs 2A and 2B and 4A and 4B), mucus hypersecretion (Fig 4C and 4D), Th2 cytokine production (Fig 3A–3D), Ag-specific cell proliferation (Fig 3E) and IgE levels in serum (Fig 2C) in WT mice. The inhibition of all these parameters was comparable in the Tim-3-/- mice.

Furthermore, the number of CD4+ and CD4+Foxp3+ Treg cells was comparable between WT and Tim-3-/- mice (S5 Fig). In our setting, respiratory tolerance is mediated via Foxp3 + Treg cells, as demonstrated in a previous paper of our group [7]. Busse et al. could show that CD4+CD25+Foxp3+ Treg numbers were increased in both spleen and lung after i.n. OVA tolerization. Furthermore Foxp3+ Treg cells isolated from tolerized WT mice and cultured in the presence of Dynabeads Mouse CD3/CD28 T Cell Expander® can secrete IL-10. They demonstrated by adoptive cell transfer experiment that these isolated Tregs conferred allergic asthma protection in our model. Though not statistically significant, the frequency of CD8+ T cells was slightly increased in all Tim-3-/- mice compared to WT animals. Furthermore, the frequency of CD4+CD154+ OVA-specific splenocytes in both WT as well as Tim-3-/- mice was reduced to approximately 3% of CD154+ cells compared to 7% in OVA-immunized mice. These OVA-specific cells of *TOL* mice produced more cytokines than *Alum* mice but less than OVA-challenged animals. In *TOL* mice the amount of Th2 cytokine producers was approximately two thirds compared to 75% in *OVA* mice and about 50% in *Alum* mice.

Taken together, we found similar allergic and tolerant phenotypes in WT and Tim-3-/- mice demonstrating that Tim-3 is not crucial in these settings.

## Discussion

To date several studies have demonstrated involvement of Tim-3 in a Th2 mediated immune response: an increased Tim-3 expression was found among CD4+ and CD4+CD25+ T cells during allergic inflammation in mice [30, 42]. These results have also been confirmed in humans among patients with asthma [32, 33]. Additionally, the number of lymphocytes was increased in Tim-3-/- mice after chronic exposure to house dust mite antigen [31]. Tim-3 was also involved in innate immune responses by upregulation of CD11c+ myeloid cells (dendritic cells, alveolar macrophages) in a murine model of allergic asthma [30]. Moreover, application of anti-Tim-3 antibody lead to a significant reduction of all parameters of airway hyperreactivity in an OVA-based model with a significant reduction in IL-5 producing cells in the lungs [28]. However, the latter murine model used by Kearley et al. was based on cell transfer of OVA-reactive Th2 cells to induce pulmonary inflammation, rather than induction by local (i.n.) or systemic (i.p.) immunization of mice with an unperturbed T cell receptor repertoire. Additionally, instead of Tim-3 deficient mice anti-Tim-3 antibody functioned as blockage of Th2 response.

In contrast, we demonstrate that Tim-3 is dispensable for the induction and maintenance of allergic airway inflammation and respiratory tolerance in a murine model. In our study, OVA-induced allergic inflammation can be equally evoked in both, WT and Tim-3 knock-out mice. We found comparably increased BALF total and differential cell counts as well as serum IgE and Th2 cytokines in both OVA-treated groups. Additionally, histological examination did not show any differences between WT and Tim-3-/- mice. These observations are in line with studies by Barlow et al. and Hiraishi et al. [30, 31]: both stated that Tim-3 is not essential for a type-2 driven allergic response in different murine models applying either OVA or house dust mite antigens in Balb/c and C57BL/6J mice, respectively ]30,31]. Our data confirm and extend these findings since we show similar frequencies and numbers of antigen-specific T

cells, similar frequencies of CD4$^+$Foxp3$^+$ Treg cells, as well as similar cytokine profiles between WT and Tim-3$^{-/-}$ mice regardless of challenge or tolerization with the model allergen OVA.

In addition to that, our study extensively investigates the induction and maintenance of respiratory tolerance in the absence of Tim-3. Previously, the putative role of Tim-3 in immunological tolerance and autoimmune diseases has been characterized in several Th1 mediated animal models: In EAE, anti-Tim-3 treatment resulted in a peracute onset of disease potentially mediated via cross-linking differentiated Th1 cells and increasing their migration which in turn led to enhanced cytokine production and macrophage activation [13]. Lee et al. extended these findings by demonstrating that blockage of Tim-3 signalling resulted in a variable disease outcome depending on different T effector cells involvement [18]. Furthermore, transplantation and peripheral tolerance was shown to be critically dependent on an undisturbed Tim-3 function [43]. Whether galectin-9 or CEACAM1 as Tim-3 ligands play essential interactive roles is still discussed controversially [44–47]. The putative role of Tim-3 for Treg cell function has been characterized by Wang *et al.* who described the influence of the Tim-3-Galectin-9 pathway on Treg cells *in vitro* and demonstrated that Tim-3 blockade inhibits Treg mediated allograft survival *in vivo* [48]. Additionally, it has been proposed by Moorman and co-workers that Tim-3 up-regulation on CD4$^+$CD25$^+$Foxp3$^+$ Treg cells correlates with Treg proliferation and suppressive function thereby modulates the Treg and T effector cell balance [49]. Recently, Gautron further characterized Tim-3$^+$ Treg cells as Th17 suppressive cells via STAT-3 expression [50].

We were therefore interested whether lack of Tim-3 has a negative impact on the induction of respiratory tolerance in our murine asthma model. Based on the previous literature about a critical role of Tim-3 on Treg function we expected that induction of respiratory tolerance in Tim-3$^{-/-}$ mice might be disabled. However, we observed decreased IL-10 levels in cell culture of spleenocytes of tolerized mice (Fig 3B) for both Tim-3$^{-/-}$ and WT mice. For Tim-4, Ahlbacker *et al.* were able to show that blockade of Tim-4 via inhibition of Tim-4 expressing lymph node medullary macrophages, resulted in an increased number of antigen-specific T cells and subsequent loss of respiratory tolerance [51]. For the first time, we characterized the role of Tim-3 in the context of a respiratory tolerance model in mice: We found comparably decreased BALF total cell numbers and eosinophils in *TOL* mice, both WT and Tim-3$^{-/-}$, compared to *OVA* mice. Moreover, Th2 cytokines (IL-4, IL-5, IL-10, IL-13), serum IgE levels and *ex vivo* proliferation of OVA-specific restimulated splenocytes were significantly diminished in both tolerized groups compared to *OVA* groups. Accordingly, IFN-γ levels were comparably increased in WT and Tim-3$^{-/-}$ *TOL* mice. Also the frequency of CD4$^+$Foxp3$^+$ Treg cells was comparable in both, WT and Tim-3$^{-/-}$ tolerized mice. OVA-specific CD4$^+$CD154$^+$ cells of TOL mice produced less cytokines than allergen-challenged animals and Th2 cytokine producers were reduced to two thirds compared to OVA and Alum mice. All these findings indicate that even Tim-3 deficient mice can be efficiently tolerized by OVA pre-treatment prior to sensitization.

Given the diverse picture of Tim-3 in innate and adaptive immunity, we cannot rule out that different mechanistic settings will lead to different findings. However, we conclude that Tim-3 is not only dispensable for the induction and maintenance of allergic airway inflammation but also for induction of respiratory tolerance in mice applying an OVA-based model. Further studies might therefore be necessary to obtain a comprehensive characterization of the role of Tim-3 in immunological tolerance.

## Supporting information

**S1 Fig. BALF differential cell count.** (A-C) BALF differential cell count was determined as number of macrophages (A), lymphocytes (B) and neutrophils (C) on cytospins. No significant

differences were detected between WT and Tim-3$^{-/-}$ *OVA* and *TOL* mice, resulting in comparable allergic and tolerant phenotypes. (Data of three independent experiments; n = 12-15 animals per group.) Mann-Whitney-*U*-test. * $p \leq 0.05$; ** $p \leq 0.01$. Data are presented as mean ± SEM.
(TIF)

**S2 Fig. OVA-specific serum immunoglobulins IgG$_1$ and IgG$_{2a}$.** (A, B) OVA-specific IgG$_1$ (A) and IgG$_{2a}$ (B) serum levels in WT and Tim-3$^{-/-}$ mice were not significantly altered in allergic vs. tolerant animals. No difference in WT vs. Tim-3$^{-/-}$ mice. (One representative out of three independent experiments; n = 5-6 animals for each group.) Mann-Whitney-*U*-test. * $p \leq 0.05$; ** $p \leq 0.01$. Data are presented as mean ± SEM.
(TIF)

**S3 Fig. Functional T cell subsets.** (A, B) Progressive gating strategy for identification of functional T cell subsets. After in vitro restimulation, splenocytes from WT (A) and Tim-3$^{-/-}$ (B) mice were incubated with amine-reactive viability dye to identify dead cells, followed by staining for CD4, fixation and intracellular staining for CD154, IFNγ, IL-17A, IL-4 and IL-5. Cells were gated through a singlet cell gate, followed by a lymphocyte gate and dead cell exclusion. T helper cells were identified using CD4 and OVA-specific T-cells were then gated using CD154 expression. The latter were then analyzed for cytokine production showing IL-4 and IL-5 here.
(TIF)

**S4 Fig. Functional composition of the CD4$^+$ T cell response.** CD4$^+$ cells were examined flow cytometrically for their production of IFNg, IL-4, IL-5, and IL-17A. The larger pie charts (light blue) show the percentage of CD4$^+$ cells that produce any of the cytokines investigated. The smaller pie charts summarize the fractions of the total response that are single-producers of any of the individual cytokines or double producers that produce two cytokines simultaneously (e.g. IL-4/IL-5 or IFNg/IL-17). For example 7,21% of the CD4$^+$ cells from WT *OVA* mice produced at least one of the investigated cytokines upon recognizing OVA. Of those, the majority was single producers of either IL-4 or IL-5 and only a small fraction produced IL-17. Responses are grouped and color-coded according to the number of functions.
(TIF)

**S5 Fig. Frequency of CD8$^+$, CD4$^+$, Foxp3$^+$ T cells.** (A, B) Splenic single-cell suspensions were first flow cytometrically examined for their percentage of CD4$^+$ (A) and CD8$^+$ T cells (B). (C) After, intracellular staining of Foxp3 was performed. No statistically significant difference was detected between the percentage of different T cell subsets in OVA immunized and challenged mice compared to their non-allergic controls. In addition, Tim-3$^{-/-}$ mice did not differ in their composition of CD8$^+$, CD4$^+$ or CD4$^+$Foxp3$^+$ T cells. Mann-Whitney-U-test. ns $p > 0.05$.
(TIF)

## Author Contributions

**Conceptualization:** Sebastian Drube, Thomas Kamradt, Gesine Hansen.

**Data curation:** Carolin Boehne, Ann-Kathrin Behrendt, Martin Boettcher.

**Formal analysis:** Carolin Boehne, Ann-Kathrin Behrendt, Martin Boettcher.

**Methodology:** Ann-Kathrin Behrendt, Almut Meyer-Bahlburg, Sebastian Drube, Thomas Kamradt, Gesine Hansen.

**Resources:** Sebastian Drube.

**Supervision:** Thomas Kamradt, Gesine Hansen.

**Validation:** Gesine Hansen.

**Visualization:** Carolin Boehne, Martin Boettcher.

**Writing – original draft:** Carolin Boehne.

**Writing – review & editing:** Carolin Boehne, Ann-Kathrin Behrendt, Almut Meyer-Bahlburg, Martin Boettcher, Sebastian Drube, Thomas Kamradt, Gesine Hansen.

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
