## [Decision Letter · Decision Letter 0]

24 Dec 2020

PONE-D-20-37656

Role of Tim-3 in experimental allergic inflammation and respiratory tolerance

PLOS ONE

Dear Dr. Boehne,

Thank you for submitting your manuscript to PLOS ONE. After careful consideration, we feel that it has merit but does not fully meet PLOS ONE’s publication criteria as it currently stands. Therefore, we invite you to submit a revised version of the manuscript that addresses the points raised during the review process.

Address reviewer's comments in revised manuscript.

We look forward to receiving your revised manuscript.

Kind regards,

Svetlana P. Chapoval

Academic Editor

PLOS ONE

Journal Requirements:

2.) In your Methods section, please provide additional information on the animal research and ensure you have included details on : (i) methods of sacrifice (ii) methods of anesthesia and/or analgesia, and (iii) efforts to alleviate suffering.

3.) Please upload a new copy of Figures S1 and S5 as the detail is not clear. Please follow the link for more information: https://blogs.plos.org/plos/2019/06/looking-good-tips-for-creating-your-plos-figures-graphics/

Additional Editor Comments:

figures should be of publication quality, please revise and submit new files.

Reviewers' comments:

Reviewer's Responses to Questions

**Comments to the Author**

1. Is the manuscript technically sound, and do the data support the conclusions?

Reviewer #1: Yes

2. Has the statistical analysis been performed appropriately and rigorously? 

Reviewer #1: Yes

3. Have the authors made all data underlying the findings in their manuscript fully available?

Reviewer #1: Yes

4. Is the manuscript presented in an intelligible fashion and written in standard English?

Reviewer #1: Yes

5. Review Comments to the Author

Reviewer #1: This is a rather straightforward study that examines the impact of Tim-3 deficiency in mice on models of allergic asthma. It appears that the mouse line used is a germ line Tim-3 KO, so thus the work addresses the overall effects of total Tim-3 deficiency on the particular in vivo models. The overall finding here is that Tim-3 is not required for development of allergic airway inflammation, at least in response to the model antigen OVA, plus alum. Use of the the model antigen, rather than an actual allergen is a caveat here. In any case, the authors also use a tolerance-promoting regimen and again show that Tim-3 is not required for induction of tolerance to OVA in the airways. These analyses, in both the inflammatory and tolerance models, were also extended to examine the production of Th1 and Th2 cytokine. Although the results here are largely negative, this study is a nice addition to the literature and appropriate for this journal. There are a few things that I would like the authors to address:

1. It is not at all clear from the manuscript how the Tim-3 KO mice were generated. Which background? Which exons(s) knocked out?

2. The authors should comment on the production of IL-10 by Treg vs. Tr1 cells; which is the more important cell type in this model? On a related note, do Tr1 cells express Tim-3?

3. As stated above, OVA is not a natural allergen, so the authors should probably soften the global conclusion which suggests that Tim-3 is not required for allergic airway inflammation writ large.

4. The figures are difficult to read, even the downloadable TIFF files, so I would like to see the figures at the proper resolution

6. PLOS authors have the option to publish the peer review history of their article (what does this mean?). If published, this will include your full peer review and any attached files.

Reviewer #1: No

---

## [Author Response · Author response to Decision Letter 0]

16 Mar 2021

1. Response to Editor

Additional Editor Comments:

figures should be of publication quality, please revise and submit new files.

We uploaded new copies of all figures. All figures were approved by PACE tool. Please let us know if there are still any other concerns about the quality of the figures.

2. Response to Reviewer

Reviewers' comments:

Reviewer's Responses to Questions

Comments to the Author

Reviewer #1: This is a rather straightforward study that examines the impact of Tim-3 deficiency in mice on models of allergic asthma. It appears that the mouse line used is a germ line Tim-3 KO, so thus the work addresses the overall effects of total Tim-3 deficiency on the particular in vivo models. The overall finding here is that Tim-3 is not required for development of allergic airway inflammation, at least in response to the model antigen OVA, plus alum. Use of the the model antigen, rather than an actual allergen is a caveat here. In any case, the authors also use a tolerance-promoting regimen and again show that Tim-3 is not required for induction of tolerance to OVA in the airways. These analyses, in both the inflammatory and tolerance models, were also extended to examine the production of Th1 and Th2 cytokine. Although the results here are largely negative, this study is a nice addition to the literature and appropriate for this journal. There are a few things that I would like the authors to address:

1. It is not at all clear from the manuscript how the Tim-3 KO mice were generated. Which background? Which exons(s) knocked out?

We added a thorough description of the generation of Tim-3-/- mice including supplemental literature (page 5, lines 94-98, (1)). In addition, Dr. Sebastian Drube who was responsible for the generation and breeding of mice was included in the list of authors. 

2. The authors should comment on the production of IL-10 by Treg vs. Tr1 cells; which is the more important cell type in this model? On a related note, do Tr1 cells express Tim-3?

IL-10 is a cytokine secreted by a wide variety of cell types including DCs, mast cells, Th2 as well as Treg cells, and as pointed out by the reviewer specific “Tr1” cells. The expression of IL-10 is highly dynamic and tightly regulated. IL-10 exerts pleiotropic stimulatory and suppressive functionalities depending on the type of immune response analyzed in vitro and in in vivo model systems. To date, IL-10 is considered to have dual functions, not only as a prototypical anti-inflammatory cytokine but also as a stimulatory cytokine promoting immune responses, e.g. by supporting B-cell and CD8+ T-cell activation (2). As mentioned by the reviewer IL-10 is also secreted by “Tr1”cells (T regulatory type 1 cells, CD4+ foxp3-) that have been reported in the literature to be induced IL-10 dependently via DC-10 cells (3). Experimental studies by Matsuda and Nabe et al. have proven, that IL-10 secreting Foxp3- Tr1 cells generated in vitro or induced in vivo by SCIT (subcutaneous immunotherapy) are able to suppress allergic inflammation (4-6). So far, the in vivo expression pattern of these cells is poorly defined, besides biomarkers like CD49b or LAG-3 (7). To date, only limited data have been published on the expression of the inhibitory receptor Tim-3. White and Wraith et al. reported that human CD4(+)IL-10(+) T cells isolated ex vivo, following a short stimulation and cytokine secretion assay, contained significantly higher proportions of TIM-3(+) and PD-1(+) cells (7). 

Up to date, we ourselves did not examine the Tr1 cell type in our OVA based allergic asthma mouse model, thus we can only generally comment on the reviewers question regarding IL-10 cytokine analysis:

We and others have shown that IL-10 is measurable in asthma or allergic disease mouse models (8,9), and observed that elevated IL-10 levels are associated with pronounced Th2 cell immune response after OVA immunization. On the other hand, experimental studies also reported that IL-10 can act suppressively as kind of a regulatory cytokine and controls Th2 cytokine secretion and eosinophilic inflammation. Furthermore, it is able to inhibit the synthesis of Th1 associated or pro-inflammatory cytokines. To date, it is known, that both Th1 and Th2 effector cells are capable to produce IL-10 after activation, this might be partially explained as a strategy of immune cells to limit their own activity and protect from damage caused by exaggerated inflammation (2,9-11). 

Our present study did not focus on elucidating the exact role of the demonstrated elevated IL-10 levels after OVA immunization. Therefore, the increase of IL-10 might by a sign of effective Th2 cell immune stimulation or even in part sign of a counterbalancing cell activity. We did not clarify this further, based on the overall similar immune response of Tim3-/- and WT mice. We observed decreased IL-10 levels in cell culture of splenocytes of tolerized mice (TOL group, Fig 3B) for both Tim-3-/- and WT. In our setting, respiratory tolerance is rather mediated via Foxp3+ Treg cells, as demonstrated in a previous paper of our group (12). Busse et al. could show that CD4+CD25+Foxp3+ Treg numbers were increased in both spleen and lung after our i.n. OVA tolerization protocol and that Foxp3+ Treg cells isolated from TOL mice cultured in the presence of Dynabeads Mouse CD3/CD28 T Cell Expander® can secrete IL-10. Busse et al. demonstrated by adoptive cell transfer experiment that these isolated Tregs conferred allergic asthma protection in our model. 

We now added supporting information on the mechanism of tolerance induction in our model to highlight the role of Treg cells involved here (page 14-15, lines 322-328). 

We apologize that we are not able to integrate new Tr1 or IL-10 related measurements into the present paper to clarify the reviewer question. However, the role of Tr1 cells might be of interest for future projects and we thank the reviewer for this valuable comment. 

3. As stated above, OVA is not a natural allergen, so the authors should probably soften the global conclusion which suggests that Tim-3 is not required for allergic airway inflammation writ large.

To further emphasize limitations given by the usage of the allergen ovalbumin, we rephrased our conclusion on the significance of Tim-3 for an OVA-based model of allergic airway disease. We changed Abstract and Discussion sections correspondingly (Abstract, page 2, line 42; Discussion, page 18, line 412).

4. The figures are difficult to read, even the downloadable TIFF files, so I would like to see the figures at the proper resolution

Thank you very much for this valuable comment. We uploaded new figure files with a better resolution. All figures were approved by PACE tool. Please let us know if there are still any other concerns about the quality of the figures.

References

 (1) Domenig C, Coyle AJ, Gutiérrez-Ramos J, Manlongat N, Kuchroo VK, Sánchez-Fueyo A, et al. Tim-3 inhibits T helper type 1-mediated auto- and alloimmune responses and promotes immunological tolerance. Nature Immunology 2003 Nov;4(11):1093-1101.

(2) Bedke T, Muscate F, Soukou S, Gagliani N, Huber S. IL-10-producing T cells and their dual functions. Seminars in immunology 2019 Aug;44:101335.

(3) Gregori S, Tomasoni D, Pacciani V, Scirpoli M, Battaglia M, Magnani CF, et al. Differentiation of type 1 T regulatory cells (Tr1) by tolerogenic DC-10 requires the IL-10–dependent ILT4/HLA-G pathway. Blood 2010 Aug 12,;116(6):935-944.

(4) Matsuda M, Doi K, Tsutsumi T, Inaba M, Hamaguchi J, Terada T, et al. Adoptive transfer of type 1 regulatory T cells suppressed the development of airway hyperresponsiveness in ovalbumin-induced airway inflammation model mice. Journal of pharmacological sciences 2019 Dec;141(4):139-145.

(5) Matsuda M, Morie Y, Oze H, Doi K, Tsutsumi T, Hamaguchi J, et al. Phenotype analyses of IL-10-producing Foxp3.sup.- CD4.sup.+ T cells increased by subcutaneous immunotherapy in allergic airway inflammation. International immunopharmacology 2018 Aug 1,;61:297.

(6) Matsuda M, Doi K, Tsutsumi T, Fujii S, Kishima M, Nishimura K, et al. Regulation of allergic airway inflammation by adoptive transfer of CD4+ T cells preferentially producing IL-10. European journal of pharmacology 2017 Oct 5,;812:38-47.

(7) White AM, Wraith DC. Tr1-like T cells – an enigmatic regulatory T cell lineage. Frontiers in immunology 2016;7:355.

(8) Polte T, Fuchs L, Behrendt A, Hansen G. Different role of CD30 in the development of acute and chronic airway inflammation in a murine asthma model. European journal of immunology 2009 Jul;39(7):1736-1742.

(9) Laouini D, Alenius H, Bryce P, Oettgen H, Tsitsikov E, Geha RS. IL-10 is critical for Th2 responses in a murine model of allergic dermatitis. The Journal of clinical investigation 2003 Oct 1,;112(7):1058-1066.

(10) Kosaka S, Tamauchi H, Terashima M, Maruyama H, Habu S, Kitasato H. IL-10 controls Th2-type cytokine production and eosinophil infiltration in a mouse model of allergic airway inflammation. Immunobiology (1979) 2011;216(7):811-820.

(11) Schülke S. Induction of Interleukin-10 Producing Dendritic Cells As a Tool to Suppress Allergen-Specific T Helper 2 Responses. Frontiers in immunology 2018;9:455.

(12) Busse M, Krech M, Meyer-Bahlburg A, Hennig C, Hansen G. ICOS Mediates the Generation and Function of CD4+CD25+Foxp3+ Regulatory T Cells Conveying Respiratory Tolerance. Journal of immunology (Baltimore, Md. : 1950) 2012 Aug 15,;189(4):1975-1982.

---

## [Editor Report · Decision Letter 1]

22 Mar 2021

Role of Tim-3 in experimental allergic inflammation and respiratory tolerance

PONE-D-20-37656R1

Dear Dr. Carolin Boehne,

We’re pleased to inform you that your manuscript has been judged scientifically suitable for publication and will be formally accepted for publication once it meets all outstanding technical requirements.

Kind regards,

Svetlana P. Chapoval

Academic Editor

PLOS ONE

Additional Editor Comments (optional):

Recommended change of title to " Tim-3 is dispensable for allergic inflammation and respiratory tolerance in experimental asthma"

Line 311 should be corrected, mice could not be dispensable for murine model
---

## [Editor Report · Acceptance letter]

29 Mar 2021

PONE-D-20-37656R1 

Tim-3 is dispensable for allergic inflammation and respiratory tolerance in experimental asthma 

Dear Dr. Boehne:

I'm pleased to inform you that your manuscript has been deemed suitable for publication in PLOS ONE. Congratulations! Your manuscript is now with our production department. 

Kind regards, 

on behalf of

Dr. Svetlana P. Chapoval 

Academic Editor

PLOS ONE